# Watch Your Step:
# Learning Node Embeddings via Graph Attention

**Sami Abu-El-Haija**[*]
Information Sciences Institute,
University of Southern California
haija@isi.edu

**Bryan Perozzi**
Google AI
New York City, NY
bperozzi@acm.org

**Rami Al-Rfou**
Google AI
Mountain View, CA
rmyeid@google.com

**Alex Alemi**
Google AI
Mountain View, CA
alemi@google.com

## Abstract

Graph embedding methods represent nodes in a continuous vector space, preserving different types of relational information from the graph. There are many hyper-parameters to these methods (e.g. the length of a random walk) which have to be manually tuned for every graph. In this paper, we replace previously fixed hyper-parameters with trainable ones that we automatically learn via backpropagation. In particular, we propose a novel attention model on the power series of the transition matrix, which guides the random walk to optimize an upstream objective. Unlike previous approaches to attention models, the method that we propose utilizes attention parameters exclusively on the data itself (e.g. on the random walk), and are not used by the model for inference. We experiment on link prediction tasks, as we aim to produce embeddings that best-preserve the graph structure, generalizing to unseen information. We improve state-of-the-art results on a comprehensive suite of real-world graph datasets including social, collaboration, and biological networks, where we observe that our graph attention model can reduce the error by up to 20%-40%. We show that our automatically-learned attention parameters can vary significantly per graph, and correspond to the optimal choice of hyper-parameter if we manually tune existing methods.

## 1 Introduction

Unsupervised graph embedding methods seek to learn representations that encode the graph structure. These embeddings have demonstrated outstanding performance on a number of tasks including node classification [29, 15], knowledge-base completion [24], semi-supervised learning [37], and link prediction [2]. In general, as introduced by Perozzi et al [29], these methods operate in two discrete steps: First, they sample pair-wise relationships from the graph through random walks and counting node co-occurances. Second, they train an embedding model e.g. using Skipgram of word2vec [25], to learn representations that encode pairwise node similarities.

While such methods have demonstrated positive results on a number of tasks, their performance can significantly vary based on the setting of their hyper-parameters. For example, [29] observed that the quality of learned representations is dependent on the length of the random walk $(C)$. In practice, DeepWalk [29] and many of its extensions [e.g. 15] use word2vec implementations [25].

---

[*]Work done while at Google AI.

Accordingly, it has been revealed by [21] that the hyper-parameter $C$, refered to as *training window length* in word2vec [25], actually controls more than a fixed length of the random walk. Instead, it parameterizes a function, we term *the context distribution* and denote $Q$, which controls the probability of sampling a node-pair when visited within a specific distance [2]. Implicitly, the choices of $C$ and $Q$, create a weight mass on every node's neighborhood. In general, the weight is higher on nearby nodes, but the specific form of the mass function is determined by the aforementioned hyper-parameters. In this work, we aim to replace these hyper-parameters with trainable parameters, so that they can be automatically learned for each graph. To do so, we pose graph embedding as end-to-end learning, where the (discrete) two steps of random walk co-occurance sampling, followed by representation learning, are joint using a closed-form expectation over the graph adjacency matrix.

Our inspiration comes from the successful application of attention models in domains such as Natural Language Processing (NLP) [e.g. 4, 38], image recognition [26], and detecting rare events in videos [31]. To the best of our knowledge, the approach we propose is significantly different from the standard application of attention models. Instead of using attention parameters to guide the model where to look when making a prediction, we use attention parameters to guide our learning algorithm to focus on parts of the data that are most helpful for optimizing an upstream objective.

We show the mathematical equivalence between the context distribution and the co-efficients of power series of the transition matrix. This allows us to learn the context distribution by learning an attention model on the power series. The attention parameters "guide" the random walk, by allowing it to focus more on short- or long-term dependencies, as best suited for the graph, while optimizing an upstream objective. To the best of our knowledge, this work is the first application of attention methods to graph embedding.

Specifically, our contributions are the following:

1. We propose an extendible family of graph attention models that can learn arbitrary (e.g. non-monotonic) context distributions.

2. We show that the optimal choice of context distribution hyper-parameters for competing methods, found by manual tuning, agrees with our automatically-found attention parameters.

3. We evaluate on a number of challenging link prediction tasks comprised of real world datasets, including social, collaboration, and biological networks. Experiments show we substantially improve on our baselines, reducing link-prediction error by 20%-40%.

## 2 Preliminaries

### 2.1 Graph Embeddings

Given an unweighted graph $G = (V, E)$, its (sparse) adjacency matrix $\mathbf{A} \in \{0, 1\}^{|V| \times |V|}$ can be constructed according to $A_{vu} = \mathbb{1}[(v, u) \in E]$, where the indicator function $\mathbb{1}[.]$ evaluates to 1 iff its boolean argument is true. In general, graph embedding methods minimize an objective:

$$\min_{\mathbf{Y}} \mathcal{L}(f(\mathbf{A}), g(\mathbf{Y}));  \tag{1}$$

where $\mathbf{Y} \in \mathbb{R}^{|V| \times d}$ is a $d$-dimensional node embedding dictionary; $f : \mathbb{R}^{|V| \times |V|} \to \mathbb{R}^{|V| \times |V|}$ is a transformation of the adjacency matrix; $g : \mathbb{R}^{|V| \times d} \to \mathbb{R}^{|V| \times |V|}$ is a pairwise edge function; and $\mathcal{L} : \mathbb{R}^{|V| \times |V|} \times \mathbb{R}^{|V| \times |V|} \to \mathbb{R}$ is a loss function.

Many popular embedding methods can be viewed in this light. For instance, a stochastic[3] version of Singular Value Decomposition (SVD) is an embedding method, and can be cast into our framework by setting $f(\mathbf{A}) = \mathbf{A}$; decomposing $\mathbf{Y}$ into two halves, the left and right representations[4] as $\mathbf{Y} = [\mathbf{L}|\mathbf{R}]$

with $\mathbf{L}, \mathbf{R} \in \mathbb{R}^{|V| \times \frac{d}{2}}$ then setting $g$ to their outer product $g(\mathbf{Y}) = g([\mathbf{L}|\mathbf{R}]) = \mathbf{L} \times \mathbf{R}^\top$; and finally setting $\mathcal{L}$ to the Frobenius norm of the error, yielding:

$$\min_{\mathbf{L},\mathbf{R}} ||\mathbf{A} - \mathbf{L} \times \mathbf{R}^\top||_F$$

## 2.2 Learning Embeddings via Random Walks

Introduced by [29], this family of methods [incl. 15, 19, 30, 10] induce random walks along $E$ by starting from a random node $v_0 \in sample(V)$, and repeatedly sampling an edge to transition to next node as $v_{i+1} := sample(\mathcal{N}[v_i])$, where $\mathcal{N}[v_i]$ are the outgoing edges from $v_i$. The transition sequences $v_0 \to v_1 \to v_2 \to \dots$ (i.e. random walks) can then be passed to word2vec algorithm, which learns embeddings by stochastically taking every node along the sequence $v_i$, and the embedding representation of this **anchor** node $v_i$ is brought closer to the embeddings of its next neighbors, $\{v_{i+1}, v_{i+2}, \dots, v_{i+c}\}$, the **context nodes**. In practice, the context window size $c$ is sampled from a distribution e.g. uniform $\mathcal{U}\{1, C\}$ as explained in [21]. For further information on graph embedding methods see [9].

Let $\mathbf{D} \in \mathbb{R}^{|V| \times |V|}$ be the co-occurrence matrix from random walks, with each entry $D_{vu}$ containing the number of times nodes $v$ and $u$ are co-visited within context distance $c \sim \mathcal{U}\{1, C\}$, in all simulated random walks. Embedding methods utilizing random walks, can also be viewed using the framework of Eq. (1). For example, to get Node2vec [15], we can set $f(\mathbf{A}) = \mathbf{D}$, set the edge function to the embeddings outer product $g(\mathbf{Y}) = \mathbf{Y} \times \mathbf{Y}^\top$, and set the loss function to negative log likelihood of softmax, yielding:

$$\min_{\mathbf{Y}} \left[ \log Z - \sum_{v,u \in V} D_{vu}(Y_v^\top Y_u) \right], \tag{2}$$

where partition function $Z = \sum_{v,u} \exp(Y_v^\top Y_u)$ can be estimated with negative sampling [25, 15].

### 2.2.1 Graph Likelihood

A recently-proposed objective for learning embeddings is the *graph likelihood* [2]:

$$\prod_{v,u \in V} \sigma(g(\mathbf{Y})_{v,u})^{D_{vu}} (1 - \sigma(g(\mathbf{Y})_{v,u}))^{\mathbb{1}[(v,u) \notin E]}, \tag{3}$$

where $g(\mathbf{Y})_{v,u}$ is the output of the model evaluated at edge $(v, u)$, given node embeddings $\mathbf{Y}$; the activation function $\sigma(.)$ is the logistic; Maximizing the graph likelihood pushes the model score $g(\mathbf{Y})_{v,u}$ towards 1 if value $D_{vu}$ is large and pushes it towards 0 if $(v, u) \notin E$.

In our work, we minimize the negative log of Equation 3, written in our matrix notation as:

$$\min_{\mathbf{Y}} ||-\mathbf{D} \circ \log(\sigma(g(\mathbf{Y}))) - \mathbb{1}[\mathbf{A} = 0] \circ \log(1 - \sigma(g(\mathbf{Y})))||_1, \tag{4}$$

which we minimize w.r.t node embeddings $\mathbf{Y} \in \mathbb{R}^{|V| \times d}$, where $\circ$ is the Hadamard product; and the L1-norm $||.||_1$ of a matrix is the sum of its entries. The entries of this matrix are positive because $0 < \sigma(.) < 1$. Matrix $\mathbf{D} \in \mathbb{R}^{|V| \times |V|}$ can be created similar to the one described in [2], by counting node co-occurrences in simulated random walks.

## 2.3 Attention Models

We mention attention models that are most similar to ours [e.g. 26, 31, 35], where an attention function is employed to suggest positions within the input example that the classification function should *pay attention to, when making inference*. This function is used during the training phase in the forward pass and in the testing phase for prediction. The attention function and the classifier are jointly trained on an upstream objective e.g. cross entropy. In our case, the attention mechanism is only guides the learning procedure, and not used by the model for inference. Our mechanism suggests *parts of the data to focus on*, during training, as explained next.

# 3 Our Method

Following our general framework (Eq 1), we set $g(\mathbf{Y}) = g([\mathbf{L} \mid \mathbf{R}]) = \mathbf{L} \times \mathbf{R}^\top$ and $f(\mathbf{A}) = \mathbb{E}[\mathbf{D}]$, the expectation on co-occurrence matrix produced from simulated random walk. Using this closed form, we extend the the Negative Log Graph Likelihood (NLGL) loss (Eq. 4) to include attention parameters on the random walk sampling.

## 3.1 Expectation on the co-occurance matrix: $\mathbb{E}[\mathbf{D}]$

Rather than obtaining $\mathbf{D}$ by simulation of random walks and sampling co-occurances, we formulate an expectation of this sampling, as $\mathbb{E}[\mathbf{D}]$. In general. this allows us to tune sampling parameters living inside of the random walk procedure including number of steps $C$.

Let $\mathcal{T}$ be the transition matrix for a graph, which can be calculated by normalizing the rows of $\mathbf{A}$ to sum to one. This can be written as:

$$\mathcal{T} = \mathrm{diag}(\mathbf{A} \times \mathbf{1_n})^{-1} \times \mathbf{A}. \tag{5}$$

Given an initial probability distribution $p^{(0)} \in \mathbb{R}^{|V|}$ of a random surfer, it is possible to find the distribution of the surfer after one step conditioned on $p^{(0)}$ as $p^{(1)} = p^{(0)^\top} \mathcal{T}$ and after $k$ steps as $p^{(k)} = p^{(0)^\top} (\mathcal{T})^k$, where $(\mathcal{T})^k$ multiplies matrix $\mathcal{T}$ with itself $k$-times. We are interested in an analytical expression for $\mathbb{E}[\mathbf{D}]$, the expectation over co-occurrence matrix produced by simulated random walks. A closed form expression for this matrix will allow us to perform end-to-end learning.

In practice, random walk methods based on DeepWalk [29] do not use $C$ as a hard limit; instead, given walk sequence $(v_1, v_2, \dots)$, they sample $c_i \sim \mathcal{U}\{1, C\}$ separately for each anchor node $v_i$ and potential context nodes, and only keep context nodes that are within $c_i$-steps of $v_i$. In expectation then, nodes $v_{i+1}, v_{i+2}, v_{i+3}, \dots$, will appear as context for anchor node $v_i$, respectively with probabilities $1, 1 - \frac{1}{C}, 1 - \frac{2}{C}, \dots$. We can write an expectation on $\mathbf{D} \in \mathbb{R}^{|V| \times |V|}$:

$$\mathbb{E}\left[\mathbf{D}^{\mathrm{DEEPWALK}}; C\right] = \sum_{k=1}^{C} \Pr(c \geq k)\tilde{\mathbf{P}}^{(0)} (\mathcal{T})^k, \tag{6}$$

which is parametrized by the (discrete) walk length $C$; where $\Pr(c \geq k)$ indicates the probability of node with distance $k$ from anchor to be selected; and $\tilde{\mathbf{P}}^{(0)} \in \mathbb{R}^{|V| \times |V|}$ is a diagonal matrix (the initial positions matrix), with $\tilde{\mathbf{P}}^{(0)}_{vv}$ set to the number of walks starting at node $v$. Since $\Pr(c = k) = \frac{1}{C}$ for all $k = \{1, 2, \dots, C\}$, we can expand $\Pr(c \geq k) = \sum_{j=k}^{C} P(c = j)$, and re-write the expectation as:

$$\mathbb{E}\left[\mathbf{D}^{\mathrm{DEEPWALK}}; C\right] = \tilde{\mathbf{P}}^{(0)} \sum_{k=1}^{C} \left[1 - \frac{k-1}{C}\right] (\mathcal{T})^k. \tag{7}$$

Eq. (7) is derived, step-by-step, in the Appendix. We are not concerned by the exact definition of the scalar coefficient, $\left[1 - \frac{k-1}{C}\right]$, but we note that the coefficient decreases with $k$.

Instead of keeping $C$ a hyper-parameter, we want to analytically optimize it on an upstream objective. Further, we are interested to learn the co-efficients to $(\mathcal{T})^k$ instead of hand-engineering a formula.

As an aside, running the GloVe embedding algorithm [28] over the random walk sequences, in expectation, is equivalent to factorizing the co-occurance matrix: $\mathbb{E}\left[\mathbf{D}^{\mathrm{GloVe}}; C\right] = \tilde{\mathbf{P}}^{(0)} \sum_{k=1}^{C} \left[\frac{1}{k}\right] (\mathcal{T})^k$.

## 3.2 Learning the Context Distribution

We want to learn the co-efficients to $(\mathcal{T})^k$. Let the context distribution $Q$ be a $C$-dimensional vector as $Q = (Q_1, Q_2, \cdots, Q_C)$ with $Q_k \geq 0$ and $\sum_k Q_k = 1$. We assign co-efficient $Q_k$ to $(\mathcal{T})^k$. Formally, our expectation on $\mathbf{D}$ is parameterized with, and is differentiable w.r.t., $Q$:

$$\mathbb{E}[\mathbf{D}; Q_1, Q_2, \dots Q_C] = \tilde{\mathbf{P}}^{(0)} \sum_{k=1}^{C} Q_k (\mathcal{T})^k = \tilde{\mathbf{P}}^{(0)} \mathop{\mathbb{E}}_{k \sim Q} [(\mathcal{T})^k], \tag{8}$$

Training embeddings over random walk sequences, using word2vec or GloVe, respectively, are special cases of Equation 8, with $Q$ fixed apriori as $Q_k = \left[1 - \frac{k-1}{C}\right]$ or $Q_k \propto \frac{1}{k}$.

| Dataset | $|V|$ | $|E|$ | nodes | edges |
|---------|------|------|-------|-------|
| wiki-vote | 7,066 | 103,663 | users | votes |
| ego-Facebook | 4,039 | 88,234 | users | friendship |
| ca-AstroPh | 17,903 | 197,031 | researchers | co-authorship |
| ca-HepTh | 8,638 | 24,827 | researchers | co-authorship |
| PPI [33] | 3,852 | 20,881 | proteins | chemical interaction |

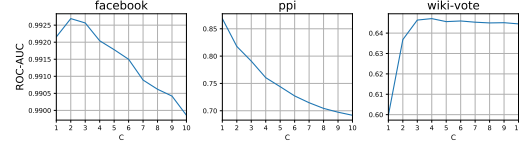

(a) Datasets used in our experiments: wiki-vote is directed but all others are undirected graphs.

(b) Test ROC-AUC as a function of C using node2vec.

Figure 1: In 1a we present statistics of our datasets. In 1b, we motivate our work by showing the necessity of setting the parameter $C$ for node2vec ($d$=128, each point is the average of 7 runs).

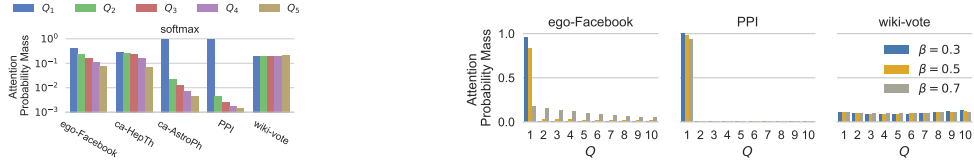

(a) Learned Attention weights $Q$ (log scale).

(b) $Q$ with varying the regularization $\beta$ (linear scale).

Figure 2: (a) shows learned attention weights $Q$, which agree with grid-search of node2vec (Figure 1b). (b) shows how varying $\beta$ affects the learned $Q$. Note that distributions can quickly tail off to zero (ego-Facebook and PPI), while other graphs (wiki-vote) contain information across distant nodes.

## 3.3 Graph Attention Models

To learn $Q$ automatically, we propose an attention model which guides the random surfer on "where to attend to" as a function of distance from the source node. Specifically, we define a *Graph Attention Model* as a process which models a node's context distribution $Q$ as the output of softmax:

$$(Q_1, Q_2, Q_3, \dots) = \text{softmax}((q_1, q_2, q_3, \dots)), \tag{9}$$

where the variables $q_k$ are trained via backpropagation, jointly while learning node embeddings. Our hypothesis is as follows. If we don't impose a specific formula on $Q = (Q_1, Q_2, \dots Q_C)$, other than (regularized) softmax, then we can use very large values of $C$ and allow every graph to learn its own form of $Q$ with its preferred sparsity and own decay form. Should the graph structure require a small $C$, then the optimization would discover a left-skewed $Q$ with all of probability mass on $\{Q_1, Q_2\}$ and $\sum_{k>2} Q_k \approx 0$. However, if according to the objective, a graph is more accurately encoded by making longer walks, then they can learn to use a large $C$ (e.g. using uniform or even right-skewed Q distribution), focusing more attention on longer distance connections in the random walk.

To this end, we propose to train softmax attention model on the infinite power series of the transition matrix. We define an expectation on our proposed random walk matrix $\mathbf{D}^{\text{softmax}[\infty]}$ as[5]:

$$\mathbb{E}\left[\mathbf{D}^{\text{softmax}[\infty]}; q_1, q_2, q_3, \dots\right] = \tilde{\mathbf{P}}^{(0)} \lim_{C \to \infty} \sum_{k=1}^{C} \text{softmax}(q_1, q_2, q_3, \dots)_k (\mathcal{T})^k, \tag{10}$$

where $q_1, q_2, \dots$ are jointly trained with the embeddings to minimize our objective.

## 3.4 Training Objective

The final training objective for the Softmax attention mechanism, coming from the NLGL Eq. (4),

$$\min_{\mathbf{L}, \mathbf{R}, \mathbf{q}} \beta \|\mathbf{q}\|_2^2 + \left\| -\mathbb{E}[\mathbf{D}; \mathbf{q}] \circ \log\left(\sigma(\mathbf{L} \times \mathbf{R}^\top)\right) - \mathbb{1}[\mathbf{A} = 0] \circ \log\left(1 - \sigma(\mathbf{L} \times \mathbf{R}^\top)\right) \right\|_1 \tag{11}$$

is minimized w.r.t attention parameter vector $\mathbf{q} = (q_1, q_2, \dots)$ and node embeddings $\mathbf{L}, \mathbf{R} \in \mathbb{R}^{|V| \times \frac{d}{2}}$. Hyper-parameter $\beta \in \mathbb{R}$ applies L2 regularization on the attention parameters. We emphasize that our attention parameters $\mathbf{q}$ live within the expectation over data $\mathbf{D}$, and are not part of the model ($\mathbf{L}, \mathbf{R}$) and are therefore not required for inference. The constraint $\sum_k Q_k = 1$, through the softmax activation, prevents $\mathbb{E}[\mathbf{D}^{\text{softmax}}]$ from collapsing into a trivial solution (zero matrix).

| Dataset | dim | Methods Use: A | | | D | | | $\mathbb{E}[\mathbf{D}]$ | Error |
| | | Eigen Maps | SVD | DNGR | n2v $C=2$ | n2v $C=5$ | Asym Proj | Graph Attention *(ours)* | Reduction |
|---|---|---|---|---|---|---|---|---|---|
| wiki-vote | 64 | 61.3 | 86.0 | 59.8 | 64.4 | 63.6 | 91.7 | $\mathbf{93.8 \pm 0.13}$ | 25.2% |
| | 128 | 62.2 | 80.8 | 55.4 | 63.7 | 64.6 | 91.7 | $\mathbf{93.8 \pm 0.05}$ | 25.2% |
| ego-Facebook | 64 | 96.4 | 96.7 | 98.1 | 99.1 | 99.0 | 97.4 | $\mathbf{99.4 \pm 0.10}$ | 33.3% |
| | 128 | 95.4 | 94.5 | 98.4 | 99.3 | 99.2 | 97.3 | $\mathbf{99.5 \pm 0.03}$ | 28.6% |
| ca-AstroPh | 64 | 82.4 | 91.1 | 93.9 | 97.4 | 96.9 | 95.7 | $\mathbf{97.9 \pm 0.21}$ | 19.2% |
| | 128 | 82.9 | 92.4 | 96.8 | 97.7 | 97.5 | 95.7 | $\mathbf{98.1 \pm 0.49}$ | 24.0% |
| ca-HepTh | 64 | 80.2 | 79.3 | 86.8 | 90.6 | 91.8 | 90.3 | $\mathbf{93.6 \pm 0.06}$ | 22.0% |
| | 128 | 81.2 | 78.0 | 89.7 | 90.1 | 92.0 | 90.3 | $\mathbf{93.9 \pm 0.05}$ | 23.8% |
| PPI | 64 | 70.7 | 75.4 | 76.7 | 79.7 | 70.6 | 82.4 | $\mathbf{89.8 \pm 1.05}$ | 43.5% |
| | 128 | 73.7 | 71.2 | 76.9 | 81.8 | 74.4 | 83.9 | $\mathbf{91.0 \pm 0.28}$ | 44.2% |

Table 1: Results on Link Prediction Datasets. Shown is the ROC-AUC. Each row shows results for one dataset results on one dataset when training embedding with We bold the highest accuracy per dataset-dimension pair, including when the highest accuracy intersects with the mean $\pm$ standard deviation. We use the train:test splits of [2], hosted on `http://sami.haija.org/graph/splits`

## 3.5 Computational Complexity

The naive computation of $(\mathcal{T})^k$ requires $k$ matrix multiplications and so is $\mathcal{O}(|V|^3 k)$. However, as most real-world adjacency matrices have an inherent low rank structure, a number of fast approximations to computing the random walk transition matrix raised to a power $k$ have been proposed [e.g. 34]. Alternatively SVD can decompose $\mathcal{T}$ as $\mathcal{T} = \mathcal{U}\Lambda\mathcal{V}^\top$ and then the $k^{\text{th}}$ power can be calculated by raising the diagonal matrix of singular values to $k$ as $(\mathcal{T})^k = \mathcal{U}(\Lambda)^k\mathcal{V}^\top$ since $\mathcal{V}^\top\mathcal{U} = I$. Furthermore, the SVD can be approximated in time linear to the number of non-zero entries [16]. Therefore, we can approximate $(\mathcal{T})^k$ in $\mathcal{O}(|E|)$. In this work, we compute $(\mathcal{T})^k$ without approximations. Our algorithm runs in seconds over the given datasets (at least 10X faster than node2vec [15], DVNE [**?** ], DNGR [8]). We leave stochastic and approximation versions of our method as future work.

## 3.6 Extensions

As presented, our proposed method can learn the weights of the context distribution $Q$. However, we briefly note that such a model can be trivially extended to learn the weight of any other type of pair-wise node similarity (e.g. Personalized PageRank, Adamic-Adar, etc). In order to do this, we can extend the definition of the context $Q$ with an additional dimension $Q_{k+1}$ for the new type of similarity, and an additional element in the softmax $q_{k+1}$ to learn a joint importance function.

# 4 Experiments

## 4.1 Link Prediction Experiments

We evaluate the quality of embeddings produced when random walks are augmented with attention, through experiments on link prediction [23]. Link prediction is a challenging task, with many real world applications in information retrieval, recommendation systems and social networks. As such, it has been used to study the properties of graph embeddings [29, 15]. Such an intrinsic evaluation emphasizes the structure-preserving properties of embedding.

Our experimental setup is designed to determine how well the embeddings produced by a method captures the topology of the graph. We measure this in the manner of [15]: remove a fraction (=50%) of graph edges, learn embeddings from the remaining edges, and measure how well the embeddings can recover those edges which have been removed. More formally, we split the graph edges $E$ into two partitions of equal size $E_{\text{train}}$ and $E_{\text{test}}$ such that the training graph is connected. We also sample non existent edges $((u, v) \notin E)$ to make $E_{\text{train}}^-$ and $E_{\text{test}}^-$. We use $(E_{\text{train}}, E_{\text{train}}^-)$ for training and model selection, and use $(E_{\text{test}}, E_{\text{test}}^-)$ to compute evaluation metrics.

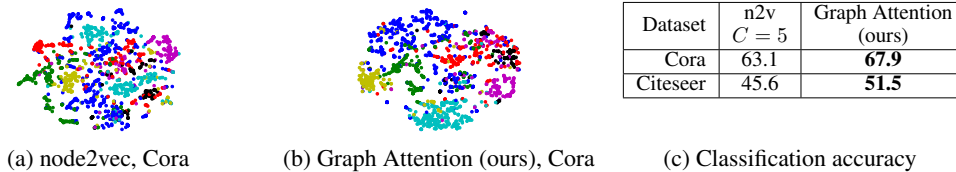

|          | (a) node2vec, Cora | (b) Graph Attention (ours), Cora | (c) Classification accuracy |
| --- | --- | --- | --- |

Figure 3: Node Classification. Fig. (a)/(b): t-SNE visualization of node embeddings for Cora dataset. We note that both methods are unsupervised, and we have colored the learned representations by node labels. Fig. (c) However, quantitatively, our embeddings achieves better separation.

**Training**: We train our models using TensorFlow, with PercentDelta optimizer [1]. For the results Table 1, we use $\beta = 0.5$, $C = 10$, and $\tilde{\mathbf{P}}^{(0)} = \text{diag}(80)$, which corresponds to 80 walks per node. We analyze our model's sensitivity in Section 4.2. To ensure repeatability of results, we have released our model and instructions[6].

**Datasets**: Table 1a describes the datasets used in our experiments. Datasets available from SNAP `https://snap.stanford.edu/data`.

**Baselines**: We evaluate against many baselines. For all methods, we calculate $g(\mathbf{Y}) \in \mathbb{R}^{|V| \times |V|}$, and extract entries from $g(\mathbf{Y})$ corresponding to positive and negative test edges, then use them to compute ROC AUC. We compare against following baselines. We mark symmetric models with †. Their counterparts, asymmetric models including ours, can learn $g(\mathbf{Y})_{vu} \neq g(\mathbf{Y})_{uv}$, which we expect to perform relatively better on the directed graph wiki-vote.
– †**EigenMaps** [5]. Minimizes Euclidean distance of adjacent nodes of $\mathbf{A}$.
– **SVD**. Singular value decomposition of $\mathbf{A}$. Inference is through the function $g(Y) = \mathcal{U}_d \times (\Lambda)_d \times \mathcal{V}_d$, where $(\mathcal{U}_d, \Lambda_d, \mathcal{V}_d)$ is a low-rank SVD decomposiiton corresponding to the $d$ largest singular values.
– †**DNGR** [8]. Non-linear (i.e. deep) embedding of nodes, using an auto-encoder on $\mathbf{A}$. We use author's code to learn the deep embeddings $\mathbf{Y}$ and use for inference $g(\mathbf{Y}) = \mathbf{Y}\mathbf{Y}^T$.
– †**n2v**: node2vec [15] is a popular baseline. It simulates random walks and uses word2vec to learn node embeddings. Minimizes objective in Eq. (2). For Table 1, we use author's code to learn embeddings $\mathbf{Y}$ then use $g(\mathbf{Y}) = \mathbf{Y}\mathbf{Y}^\top$. We run with $C = 2$ and $C = 5$.[7]
– **AsymProj** [2]. Learns edges as asymmetric projections in a deep embedding space, trained by maximizing the graph likelihood (Eq. 3).

**Results**: Our results, summarized in Table 1, show that our proposed methods substantially outperform all baseline methods. Specifically, we see that the error is reduced by up to $45\%$ over baseline methods which have fixed context definitions. This shows that by parameterizing the context distribution and allowing each graph to learn its own distribution, we can better preserve the graph structure (and thereby better predict missing edges).

**Discussion**: Figure 2a shows how the learned attention weights $Q$ vary across datasets. Each dataset learns its own attention form, and the highest weights generally correspond to the highest weights when doing a grid search over $C$ for node2vec (as in Figure 1b).

The hyper-parameter $C$ determines the highest power of the transition matrix, and hence the maximum context size available to the attention model. We suggest using large values for $C$, since the attention weights can effectively use a subset of the transition matrix powers. For example, if a network needs only 2 hops to be accurately represented, then it is possible for the softmax attention model to learn $Q_3, Q_4, \cdots \approx 0$. Figure 2b shows how varying the regularization term $\beta$ allows the softmax attention model to "attend to" only what each dataset requires. We observe that for most graphs, the majority of the mass gets assigned to $Q_1, Q_2$. This shows that shorter walks are more beneficial for most graphs. However, on wiki-vote, better embeddings are produced by paying attention to longer walks, as its softmax $Q$ is uniform-like, with a slight right-skew.

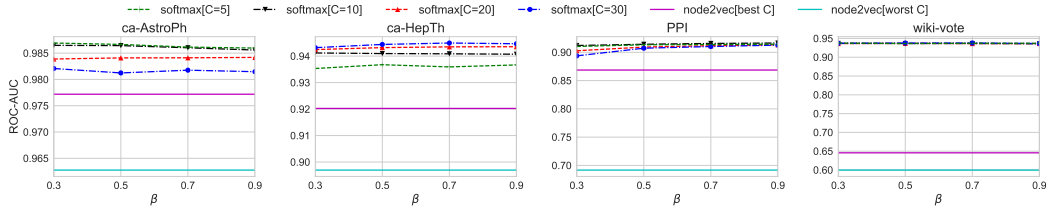

Figure 4: Sensitivity Analysis of softmax attention model. Our method is robust to choices of both $\beta$ and $C$. We note that it consistently outperforms even an optimally set node2vec.

## 4.2 Sensitivity Analysis

So far, we have removed two hyper-parameters, the maximum window size $C$, and the form of the context distribution $\mathcal{U}$. In exchange, we have introduced other hyper-parameters – specifically walk length (also $C$) and a regularization term $\beta$ for the softmax attention model. Nonetheless, we show that our method is robust to various choices of these two. Figures 2a and 2b both show that the softmax attention weights drop to almost zero if the graph can be preserved using shorter walks, which is not possible with fixed-form distributions (e.g. $\mathcal{U}$).

Figure 4 examines this relationship in more detail for $d = 128$ dimensional embeddings, sweeping our hyper-parameters $C$ and $\beta$, and comparing results to the best and worst node2vec embeddings for $C \in [1, 10]$. (Note that node2vec lines are horizontal, as they do not depend on $\beta$.) We observe that all the accuracy metrics are within $1\%$ to $2\%$, when varying these hyper-parameters, and are all still well-above our baseline (which sample from a fixed-form context distribution).

## 4.3 Node Classification Experiments

We conduct node classification experiments, on two citation datasets, Cora and Citeseer, with the following statistics: Cora contains ($2,708$ nodes, $5,429$ edges and $K = 7$ classes); and Citeseer contains ($3,327$ nodes, $4,732$ edges and $K = 6$ classes). We learn embeddings from only the graph structure (nodes and edges), without observing node features nor labels during training. Figure 3 shows t-SNE visualization of the Cora dataset, comparing our method with node2vec [15]. For classification, we follow the data splits of [37]. We predict labels $\widetilde{L} \in \mathbb{R}^{|V| \times K}$ as: $\widetilde{L} = \exp\left(\alpha g(\mathbf{Y})\right) \times L_{\text{train}}$, where $L_{\text{train}} \in \{0,1\}^{|V| \times K}$ contains rows of ones corresponding to nodes in training set and zeros elsewhere. The scalar $\alpha \in \mathbb{R}$ is manually tuned on the validation set. The classification results, summarized in Table 3c, show that our model learns a better unsupervised representation than previous methods, that can then be used for supervised tasks. We do not compare against other semi-supervised methods that utilize node features during training and inference [incl. 37, 20], as our method is unsupervised.

Our classification prediciton function contains one scalar parameter $\alpha$. It can be thought of a "smooth" k-nearest-neighbors, as it takes a weighted average of known labels, where the weights are exponential of the dot-product similarity. Such a simple function should introduce no model bias.

## 5 Related Work

The field of learning on graphs has attracted much attention lately. Here we summarize two broad classes of algorithms, and point the reader to recent reviews [10, 6, 18, 14] for more context.

The first class of algorithms are semi-supervised and concerned with predicting labels over a graph, its edges, and/or its nodes. Typically, these algorithms process a graph (nodes and edges) as well as per-node features. These include recent graph convolution methods [e.g. 27, 7, 3, 17] with spectral variants [12, 7], diffusion methods [e.g. 11, 13], including ones trained until fixed-point convergence [32, 22] and semi-supervised node classification [37] with low-rank approximation of convolution [12, 20]. We differ from these methods as (1) our algorithm is unsupervised (trained exclusively from the graph structure itself) without utilizing labels during training, and (2) we explicitly model the relationship between all node pairs.

The second class of algorithms consist of unsupervised graph embedding methods. Their primary goal is to preserve the graph structure, to create task independent representations. They explicitly model the relationship of all node pairs (e.g. as dot product of node embeddings). Some methods directly use the adjacency matrix [8, 36], and others incorporate higher order structure (e.g. from simulated random walks) [29, 15, 2]. Our work falls under this class of algorithms, where inference is a scoring function $V \times V \to \mathbb{R}$, trained to score positive edges higher than negative ones. Unlike existing methods, we do not specify a fixed context distribution apriori, whereas we push gradients through the random walk to those parameters, which we jointly train while learning the embeddings.

## 6  Conclusion

In this paper, we propose an attention mechanism for learning the context distribution used in graph embedding methods. We derive the closed-form expectation of the DeepWalk [29] co-occurrence statistics, showing an equivalence between the context distribution hyper-parameters, and the coefficients of the power series of the graph transition matrix. Then, we propose to replace the context hyper-parameters with trainable models, that we learn jointly with the embeddings on an objective that preserves the graph structure (the Negative Log Graph Likelihood, NLGL). Specifically, we propose Graph Attention Models, using a softmax to learn a free-form contexts distribution with a parameter for each type of context similarity (e.g. distance in a random walk).

We show significant improvements on link prediction and node classification over state-of-the-art baselines (that use a fixed-form context distribution), reducing error on link prediction and classification, respectively by up to 40% and 10%. In addition to improved performance (by learning distributions of arbitrary forms), our method can obviate the manual grid search over hyper-parameters: walk length and form of context distribution, which can drastically fluctuate the quality of the learned embeddings and are different for every graph. On the datasets we consider, we show that our method is robust to its hyper-parameters, as described in Section 4.2. Our visualizations of converged attention weights convey to us that some graphs (e.g. voting graphs) can be better preserved by using longer walks, while other graphs (e.g. protein-protein interaction graphs) contain more information in short dependencies and require shorter walks.

We believe that our contribution in replacing these sampling hyperparameters with a learnable context distribution is general and can be applied to many domains and modeling techniques in graph representation learning.

## Footnotes

[2]To clarify, as noted by Levy et al [21] – studying the implementation of word2vec reveals that rather than using $C$ as constant and assuming all nodes visited within distance $C$ are related, a desired context distance $c_i$ is sampled from uniform ($c_i \sim \mathcal{U}\{1, C\}$) for each node pair $i$ in training. If the node pair $i$ was visited more than $c_i$-steps apart, it is not used for training. Many DeepWalk-style methods inherited this context distribution, as they internally utilize standard word2vec implementations.

[3]Orthonormality Constraints are not shown.

[4]Also known in NLP [25] as the "input" and "output" embedding representations.

[5]We do *not* actually unroll the summation in Eq. (10) an infinite number of times. Our experiments show that unrolling it 10 or 20 times is sufficient to obtain state-of-the-art results.

[6]Available at `http://sami.haija.org/graph/context`

[7]We sweep $C$ in Figure 1b, showing that there are no good default for $C$ that works best across datasets.

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
