[Supplementary Material · appendix.pdf]

# 7 Appendix

## 7.1 Step-by-step Derivation of Equation (7)

Let $x(k)$ be the position of random surfer at time $k$. Speficically, $x : \mathbb{Z}^+ \to V$. We assume a Markov chain: The value of $x(k)$ only depends on previous step: $x(k-1)$. To calulate the expectation $\mathbb{E}[\mathbf{D}]$, the square node-to-node co-occurence matrix, we start by calculating one entry at a time: $\mathbb{E}[D_{uv}]$, the expected number of times that $u$ is selected in $v$'s context. Let $W_v(k)$ be the *context set* that gets sampled if $v$ is visited at the $k^{\text{th}}$ step. Concretely, if $x(k) = v$, and the random walker continues the sequence, $x(k+1) = v'_1$ then $x(k+2) = v'_2$ then $x(k+3) = v'_3 \ldots$, the context set of DeepWalk can be defined as $W_v(k) = \{v'_1, v'2 \ldots, v'_c$, where $c \sim \mathcal{U}\{1, C\}$. We would like to count the event $u \in W_v(k)$ for every $k \in \{1, 2, \ldots, C\}$.

Using Markov Chain, we can write:

$$\Pr\left(x(i+k) = u \mid x(i) = v\right) = \Pr\left(x(k) = u \mid x(0) = v\right)$$
$$= \left(\mathcal{T}^k\right)_{uv} \tag{12}$$

Now, if node $u$ was visited $k$ steps after node $v$, then the probabilitiy of it being sampled is given by:

$$\Pr\left(u \in W_v(k) \mid x(k) = u, x(0) = v\right). \tag{13}$$

In case of DeepWalk [29], probability above equals:

$$\Pr\left(c \geq k \mid x(k) = u, x(0) = v\right) \text{ where } c \sim \mathcal{U}\{1, C\}, \tag{14}$$

and event $k \leq c$ is independant of the condition $(x(k) = u \cap x(0) = v)$. Further, event $k \leq c$ can be partitioned and Eq. (14) can be written as

$$\Pr\left(c = k \cup c = k+1 \cup \cdots \cup c = C\right) \tag{15}$$

$$= \sum_{j=k}^{C} \Pr\left(c = j\right) \tag{16}$$

$$= (C - k + 1)\left(\frac{1}{C}\right) = 1 - \frac{k-1}{C}, \tag{17}$$

where second line is trivial since the events $c = j$ are disjoint. We can now use Bayes' rule to derive the probability of $u$ being visited $k$ steps after $v$ _and_ being selected in $v$'s sampled context, as:

$$\Pr\left(u \in W_v(k), x(k) = u \mid x(0) = v\right)$$
$$= \Pr\left(u \in W_v(k) \mid x(k) = u, x(0) = v\right) \Pr\left(x(k) = u \mid x(0) = v\right)$$
$$= \left(1 - \frac{k-1}{C}\right)\left(\mathcal{T}^k\right)_{uv} \tag{18}$$

Now, let $E_{vku}$ be the event that a walker visits $v$ and after $k$ steps, visits $u$ and selects it part of its context. This event happens with the probability indicated in Equation 18. Concretely,

$$\mathbb{E}\left[E_{vku} \mid x(0) = v\right] = \left(1 - \frac{k-1}{C}\right)\left(\mathcal{T}^k\right)_{uv}. \tag{19}$$

Let $E_{v*u}$ count the events $\{E_{vku} : k \in [1, C]\}$, then:

$$\mathbb{E}\left[E_{v*u} \mid x(0) = v\right] = \mathbb{E}\left[\sum_{k=1}^{C} E_{vku} \;\middle|\; x(0) = v\right] \tag{20}$$

$$= \sum_{k=1}^{C} \mathbb{E}\left[E_{vku} \mid x(0) = v\right] = \sum_{k=1}^{C} \left(1 - \frac{k-1}{C}\right)\left(\mathcal{T}^k\right)_{uv}. \tag{21}$$

Suppose we run DeepWalk, starting $m$ random walks from each node $v$, then the expected number of times that $u$ is present in the context of $v$ is given by:

$$\mathbb{E}\left[D_{uv}^{\text{DEEPWALK}}\right] = m\mathbb{E}\left[E_{v*u} \mid x(0) = v\right] = m\sum_{k=1}^{C} \left(1 - \frac{k-1}{C}\right)\left(\mathcal{T}^k\right)_{uv}.$$

Finally, we can write down the expectation over the square matrix $\mathbf{D}$:

$$\mathbb{E}\left[\mathbf{D}^{\text{DEEPWALK}}\right] = \text{diag}(m, m, \ldots, m) \sum_{k=1}^{C} \left(1 - \frac{k-1}{C}\right) \left(\mathcal{T}^k\right)$$

$$= \text{Equation (7)}$$

$\square$

## 7.2 Choice of $\tilde{\mathbf{P}}^{(0)}$

The github code of DeepWalk and node2vec start a fixed number ($m$) walks from every graph node $v \in V$. For node2vec, $m$ defaults to 10, see `num-walks` flag in `https://github.com/aditya-grover/node2vec/blob/master/src/main.py`. Therefore, in our experiments, we set $\tilde{\mathbf{P}}^{(0)} := \text{diag}(m, m, \ldots, m)$. This initial condition yields $D_{vu}$ to be the expected number of times that $u$ is visited if we started $m$ walks from $v$. There can be other reasonable choices. Nonetheless, we use what worked well in practice for [15, 29]. We leave the search for a better $\tilde{\mathbf{P}}^{(0)}$ as future work.

## 7.3 Depiction of Learned Context Distribution

Figure: Depiction of how our model assigns context distributions (shaded red) compared to earlier work. We depict the graph from the perspective of anchor node (yellow). Given a social graph (top), where friends of friends are usually friends, our algorithm learns a leftskewed distribution. Given a voting graph (bottom), with general transitivity: $a \rightarrow b \rightarrow c \implies a \rightarrow c$, it learns a long-tail distribution. Earlier methods (e.g. DeepWalk) use word2vec, which internally uses a linear decay context distribution, treating all graphs the same.