[Reviews · NeurIPS 2018]

Reviewer 1



The paper proposes a new algorithm for learning node embedding, by bringing together the attention model and the graph likelihood objective function suggested in a recent work [2]. By learning the context distribution determining the coefficients of powers of the transition matrix, it leads to a more flexible representation learning model. Quality: Although the technical content of the paper seems to be correct, I would like to point out some issues in the paper: - In Line 121, the sentence "In practice, random walk methods based on Deepwalk do not use $C$ as a hard limit" is not generally correct because many methods (such as Node2vec) perform fixed length walks. - Although the proposed method is described using the random walk concept, it computes the expected value of the number of node pair occurrences in a length $c \sim \mathcal{U}(1,C)$ by applying matrix multiplication. On the other hand, Deepwalk or Node2vec implicitly approximate it, by performing only a limited number of random walks. Therefore, I think it could be also better to see its performance against the methods such as HOPE. - In classification, only two datasets have been used. I would strongly recommend to examine the performance of the method over different training and test sizes. - In the Table b of Figure 1, the figures for ca-AstroPh and ca-HepTh are not depicted. Those results are important to better understand the behavior of the proposed method: in Table 1, Graph Attention uses windows size of $C=10$ but node2vec uses $C=2$ or $C=5$. - Although experimental results are given with standard deviation for the Graph Attention method, in Table 1 only mean scores are shown for the other methods. - Is the proposed method scalable for large networks? This is a very important question, since the proposed algorithm performs matrix factorization. Clarity: I think the paper is well-organized and quite readable. In the beginning, the authors shortly mention two previous works on "Attention Models" and "Graph Likelihood", and then they introduce the proposed method by incorporating ideas from the above two methodologies. However, I would like to make some suggestions for the notation used in the paper: - Between lines 29-30, the symbol $C$ is used to indicate the length of the walk and the italic $\mathcal{C}$ is also used to denote the context distribution but it can be confused with the other notation $C$. - In Line 63, the expression "$|E|$ non-zero entries" is not correct. For example the adjacency matrix of a simple graph have $2|E|$ non-zero entries due to symmetry. - In line 72, if $E[v_i]$ are the outgoing edges from $v_i$ then $sample(E[v_i])$ returns an edge and not a node -- so it could be better to define $E[v_i]$ as the neighbourhood of vertex $v_i$. - In line 121, the sentence "In practice, random walk methods based on Deepwalk do not use $C$ as a hard limit" is not generally correct because many methods such as Node2vec perform fixed length walks. Originality: The authors extend a previous work on the “graph likelihood objective” [2] with the Attention Model, and the proposed method outperforms the state-of-the-art baselines for the link prediction task. Significance: This work incorporates the attention model with the "graph likelihood" objective function proposed by a recent work [2], in such a way that it enables to learn context distribution which is used in the paper to compute the expected value of the number of co-occurrences of nodes. Although the context distribution corresponds to the coefficients of the powers of the transition matrix, I believe that the same idea can be applied for different models. The experiments for node classification appear to be insufficient to evaluate the model, but the suggested method shows significant improvements for the link prediction task.

Reviewer 2



The authors point out the importance of selecting context for most of graph embedding learning and then propose the trainable framework to properly choose the context from random walks. The idea of attention models widely used in language models is introduced to identify the important context. The experimental results show that the proposed method outperforms the baseline methods in both link prediction and node classification tasks. * Strengths - The authors tackle the very important problem of choosing context in the network and propose a reasonable solution. - The manuscript is easy to follow. - Trained models are well-studied beyond the performance on the predictive tasks. * Weakness - Despite the comparison with node2vec, which uses the second-order Markov chain for its random-walk sequence, the authors only deal with the first-order Markov chain. It means that D in nodevec is not be obtainable by the power-series in the proposed method. Despite this difference, all the random-walks are regarded the same as simple random walks. - One of the main advantages from the simulated random-walks is parallelization (at least for sequence generation) and scalability. Section 3.5 is not enough for scalability argument. * Detailed comments - Due to [28] and the attention usage in the language models, the attention is generally regarded per instance (or sentence/sequence) while here the attention is globally defined. The proposed method has its own value as the authors describe in the paper, but the terminology can lead to some confusion. - Some references such as [16] need to be polished. - Section 3.5 seems somewhat contradictory to the original purpose of trying the graph embedding because it will cut out the information that SVD cannot preserve. Also, whether Section 3.5 is just an idea or it is actually implemented to be used in experiments is not clear. It would be interesting to see the difference in performance between full-matrix context and SVD-approximate context. - Section 3.6 seems more promising to generalize the proposed idea beyond the simple random-walks. - While the authors sweep C for node2vec, the other hyperparameters are not mentioned where it is hard to believe that the default values of the other hyperparameters work best across the datasets. Mentioning the detail about C without mentioning the others naturally raise those questions, so the authors need to add some description.

Reviewer 3



This paper proposes a new model for generating Graph node embeddings without the need to manually tune relevant hyperparameters, but rather rely on learning trainable node contexts distribution through attention mechanism to model the relationship between all Graph nodes. These learned attention scores are shown to vary between different graphs, indicating that the model is actually able to preserve the inherent graph information. The paper is well written and structured, presenting the problem clearly and accurately. It contains considerable relevant references and background knowledge. It nicely motivates the proposed approach, locates the contributions in the state-of-the-art and reviews related work. It aligns specifically with the unsupervised Graph embedding methods. It is also very clear in terms of how it differs on the technical level from existing approaches. What I am missing, however, is a discussion of pros and cons. How should we improve in the future? Overall, however, the ideas presented in the paper are interesting and original, presumably applicable to multiple domains including Graph attention or Graph embedding. And the experimental section is convincing. My recommendation is to accept this submission.